

# Emergence of quasiparticle Bloch states in artificial crystals crafted atom-by-atom

Jan Girovsky[1], Jose L. Lado[2], Floris E. Kalff[1], Eleonora Fahrenfort[1],
Lucas J. J. M. Peters[1], Joaquín Fernández-Rossier[2,3] and Alexander F. Otte[1*]

**1** Department of Quantum Nanoscience, Kavli Institute of Nanoscience,
Delft University of Technology, Lorentzweg 1, 2628 CJ Delft, The Netherlands
**2** QuantaLab, International Iberian Nanotechnology Laboratory (INL),
Avenida Mestre José Veiga, 4715-310 Braga, Portugal
**3** Departamento de Física Aplicada, Universidad de Alicante, San Vicente del Raspeig,
03690 Spain

* a.f.otte@tudelft.nl

## Abstract

The interaction of electrons with a periodic potential of atoms in crystalline solids gives rise to band structure. The band structure of existing materials can be measured by photoemission spectroscopy and accurately understood in terms of the tight-binding model, however not many experimental approaches exist that allow to tailor artificial crystal lattices using a bottom-up approach. The ability to engineer and study atomically crafted designer materials by scanning tunnelling microscopy and spectroscopy (STM/STS) helps to understand the emergence of material properties. Here, we use atom manipulation of individual vacancies in a chlorine monolayer on Cu(100) to construct one- and two-dimensional structures of various densities and sizes. Local STS measurements reveal the emergence of quasiparticle bands, evidenced by standing Bloch waves, with tuneable dispersion. The experimental data are understood in terms of a tight-binding model combined with an additional broadening term that allows an estimation of the coupling to the underlying substrate.


Atom manipulation by means of STM is a viable way of constructing atomically precise artificial structures [1]. Among others, the technique can be used to engineer atomic scale logic devices [2,3], low dimensional magnetic systems [4–6] or atomic data storages [7–9]. As our abilities to manipulate atoms on a large scale are improving, the formation of atomically designed artificial crystals becomes of particular interest driven by a demand for new materials where the properties are defined by emerging quasiparticle states [10]. Common approaches to build low-dimensional artificial materials by STM include confinement of electronic surface states through precise assembly of individual atoms and/or molecules [11–14], self-assembly of molecular networks [15,16] and manipulation of dangling bonds [17] or surface vacancies [18]. The recent development of large-scale fully automated placement of atomic vacancies on a chlorinated copper crystal surface [9] provides an excellent platform to explore various lattice compositions. These vacancies were found to host a localized vacancy state in the surface band gap, similar to dopants in semiconductors, allowing their combined electronic states to be modelled by means of tight-binding approximation [19].

Here, we present a study of artificial one- and two-dimensional structures built from Cl vacancies in an otherwise perfect monolayer square lattice formed by chlorine atoms on a Cu(100) surface. Using local electron tunnelling spectroscopy, we demonstrate that we are able to reach system scales where the spectral properties no longer depend on size and which we therefore consider to be in the limit of infinite lattice size. For structures with a sufficiently large vacancy density, we observe quasiparticle Bloch waves that can be simulated by using a tight-binding model. Similar wave patterns were reported previously in assembled chains of Au atoms [14], which were best described in terms of a free electron model. Analysis of the Bloch wave dispersion allows us to extract quasiparticle effective masses, which are found to depend strongly on the chosen lattice structure.

A monolayer of chlorine atoms on Cu(100) exhibits a surface band gap $E_g$ of about 7 eV (see inset of Fig. 1a) as well as a shift in the substrate's work function by 1.25 eV [20], suggesting a significant charge transfer between the substrate and chlorine atoms and formation of the interface dipole moment [21]. Theoretical calculations predict a charge of 0.5 electron accumulated on chlorine atoms and depletion of the density of states (DOS) at the top-most layer of the copper substrate [22]. Other materials with a similarly large surface band gap, e.g. $Cu_2N$ on Cu(100) ($E_g \sim 4$ eV) [23], NaCl bilayers on copper substrates ($E_g \sim 8.5$ eV) [24], and non-polar MgO films on Ag(100) ($E_g \sim 6$ eV) [25], have found applications in studies of elementary excitations in individual molecules and/or adatoms [3, 26–28]. The insulating monolayers formed on the metal substrates have been shown to have a little effect on the valence band maximum, however significantly affect the conduction band minimum, which was found as high as $\sim 4$ eV for NaCl bi- and tri-layers on copper [24]. In our case, a sharp step in the differential conductance at $\sim 3.5$ V denotes the conduction band minimum (Fig. 1a, black curve). The precise onset of the band was determined as the maximum in the normalized differential conductance $dI/dV \times V/I$ (see Fig. 5).

As previously reported by Drost et al. [19], when the Cl/Cu(100) interface possesses defects in the form of missing chlorine atoms (dark square in the inset of Fig. 1a), a localized electronic vacancy state is resolved at lower voltages $\sim 3.4$ V (green curve Fig. 1a). The vacancy state exhibits similarities to localized states observed on gold atoms adsorbed on NiAl(110) [14], in the gap region of hydrogen-doped Si(100) surface [17], and on chlorine vacancies in NaCl/Cu(111) [24]. When two vacancies are brought close to each other by means of atom manipulation [9], the spatial overlap of the wave functions leads to the formation of bonding and anti-bonding orbitals [14, 17, 24]. These molecular orbitals can be effectively described within the tight-binding model with their energy depending on the hopping term $t$ – a measure of the overlap of the two vacancy states through the underlying potential.

We built one-dimensional lattices of the Cl/Cu(100) vacancies of various lengths and lattice spacing ({3,0}, {2,0} and {1,1}) as shown in Fig. 1. The notation $\{x, y\}$ used here describes spacing between adjacent vacancies in the horizontal and vertical directions, respectively, in multiplies of the lattice constant $a = 3.55$ Å. Differential conductance ($dI/dV$) spectra, presented in Fig. 1, were acquired along chains of length 16, for all three spacing parameters. The spectra reveal a shift of the band onset towards lower voltages and broadening of the spectral features for the lattice of denser spacing. Both the shift and the band broadening result from the increased overlap between neighbouring sites. The observed spectral features show a correlation with the position within the chains, i.e. the band minimum measured on outer vacancies is found at higher energies compared to that resolved on inner ones. The correlation of the band minimum with the position within the chain is related to the broken translational symmetry at the outer positions and leads to the appearance of zero-dimensional states [14]. This dependence is further corroborated for edge vacancies within denser lattices where the effect is more pronounced (e.g. Fig. 1e). Spectra acquired on the chlorine atoms within the chains show a similar spatial dependence of the band onset.

In Fig. 1f, we plot the dependence of the band onset as a function of the chain length, measured at the centre of each chain. For each lattice spacing, the band onset is found to saturate, however at different lengths: the $\{3,0\}$ chains saturate at 3.35 eV already for length 3, the $\{2,0\}$ chains at 3.18 eV for length 5 and the $\{1,1\}$ chains at 3.1 eV for length 8. The saturation of the band minimum implies an approach of the limit where edge effects no longer play a role for the inner vacancies and the chains can be effectively treated in the limit of infinite lattice size. Furthermore, the observed shift relates to the size of the hopping parameter $t$ which increases for shorter lattice spacing [19].

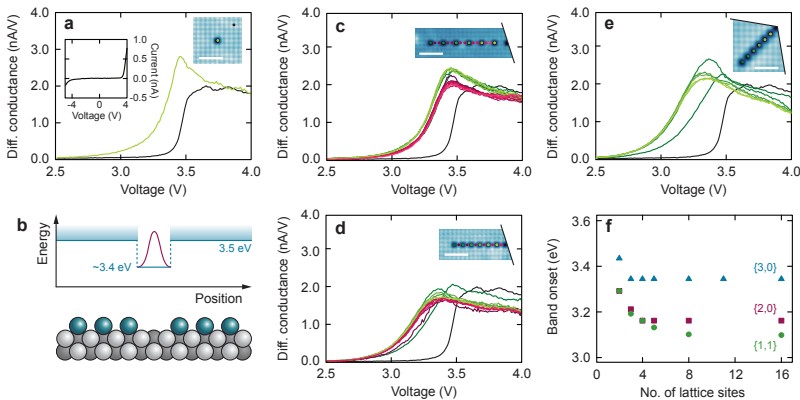

Figure 1: Differential conductance spectra acquired on a chlorine vacancy in the Cl/Cu(100) substrate and in artificial 1D lattices crafted from the vacancies. (**a**) $dI/dV$ measurement taken inside a chlorine vacancy (green) and on the bare Cl/Cu(100) substrate (black). Dots depict positions where spectra were taken. The inset shows an *I-V* curve acquired from -4.5 to 4.0 V where the surface band gap of $E_g \sim 7$ eV is clearly visible. (**b**) Sketch denoting the energy level of the vacancy state with respect to the band continuum. (**c**, **d**, **e**) $dI/dV$ spectra taken on vacancy sites and/or chlorine interstices (locations indicated by coloured dots in the insets) on lattices with 16 vacancy sites for spacing configurations $\{3,0\}$ (**c**), $\{2,0\}$ (**d**) and $\{1,1\}$ (**e**). (**f**) Evolution of the band onset as a function of lattice spacing and lattice size. Data points indicate the position of the band onset extracted from spectra taken on the middle vacancy of each lattice. In the case of an even-length chain, the averaged value measured on the two centre vacancies is shown. STM images were acquired in constant current mode $I = 2$ nA and $V = 500$ mV. All scale bars are 2 nm.

To further investigate the band formation, we built two-dimensional structures with varying lattice spacing, ($\{3,3\}$, $\{2,3\}$, $\{2,2\}$) as well as 'stripes' and 'checkerboard' arrays, all of varying lattice size (Fig. 2). For 2D lattices, the notation $\{x,y\}$ denotes the lattice spacing in the $x$ and $y$ directions in units of the lattice constant $a$. Moving inward along the diagonal of each structure, the position of the band onset shifts towards lower energies for denser and larger lattices, similar to the 1D lattices. In the case of the stripes lattice we observe two band onsets, $E_1 = 2.8$ eV and $E_2 = 3.1$ eV, measured at the centre vacancy (see Fig. 2g). We attribute these to the two different lattice constants along the lattice diagonals, $a_1 = 0.51$ nm and $a_2 = 0.69$ nm. Assuming that the hopping parameter is exponentially dependent on the distance [19] and the bandwidth is linearly proportional to the hopping parameter $t$, the band is expected to be symmetric around the energy $E = 3.4$ eV of a single vacancy. We estimate the widths of the respective bands to be $W_1 \sim 1.2$ eV and $W_2 \sim 0.6$ eV, leading to a ratio $W_1/W_2 \sim 2$. This ratio is somewhat higher than the ratio between the hopping parameters $t(a_1)/t(a_2) \sim 1.2$, suggesting that another effect may play a role, affecting the width and/or position of the band, e.g. an electric field due to positively charged neighbouring vacancies

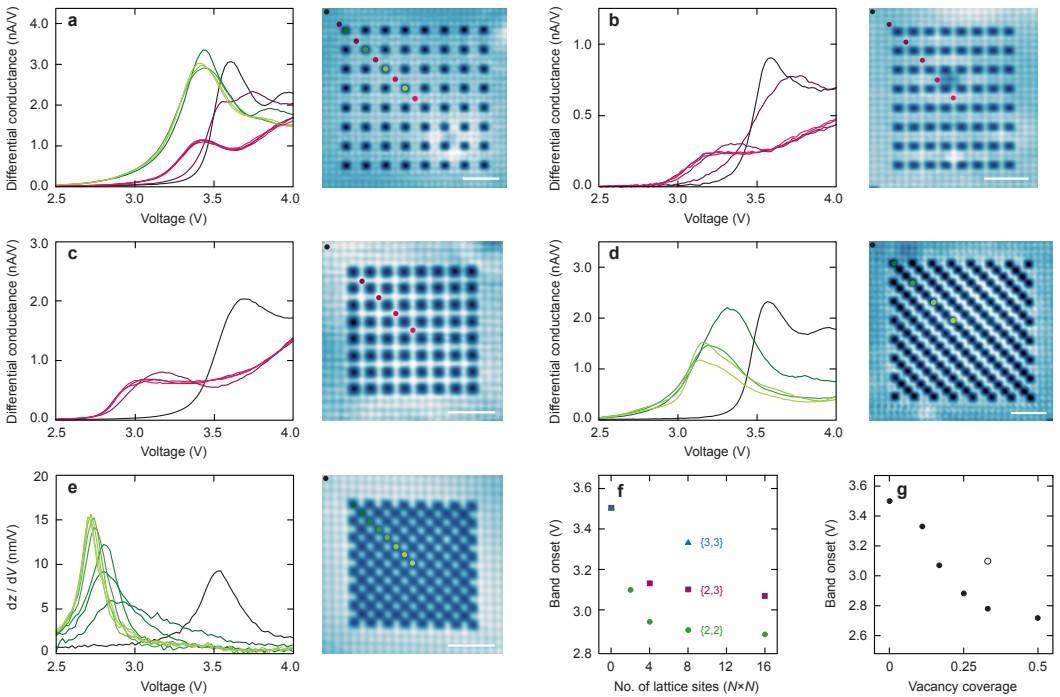

Figure 2: Analysis of artificial 2D lattices by chlorine vacancies of Cl/Cu(100). (**a**, **b**, **c**, **d**, **e**) Differential conductance spectra measured along diagonals of the lattices with spacing {3, 3} (**a**), {2, 3} (**b**), {2, 2} (**c**) and with stripes (**d**) and checkerboard (**e**) patterns. Coloured dots denote the positions where the spectra were acquired. (**f**) Evolution of band onset as a function of lattice spacing and lattice size. Band onset was extracted from the spectra taken in the middle of the lattices. (**g**) Band onset as a function of lattice density, i.e. number of vacancies divided by number of total positions in the unit cell. STM images were acquired in constant current mode $I = 2$ nA and $V = 500$ mV. All scale bars 2 nm.

observed at polar insulating surfaces. Such an electric field can cause a shift of the band onset towards lower energies, which is expected to be larger for denser lattices [18].

The checkerboard lattice (Fig. 2e) was found to be more sensitive to relatively high tunnelling currents than lattices with lower vacancy coverage. For large tunnelling current and voltage values (e.g. > 2 nA at ∼ 4.7 V), we observed chlorine atoms changing their position, rendering the structure unstable. For this reason, instead of acquiring d$I$/d$V$ spectra, we measured the dependence of the tip-sample distance $z$ as a function of sample voltage in constant current mode, i.e. d$z$/d$V$ curves, that qualitatively resemble the normalized differential conductance d$I$/d$V$×$V$/$I$ (Fig. 5).

Apart from preserving the lattice integrity, the d$z$/d$V$ measurement mode also provides sufficient sensitivity to detect standing wave modes in some of the lattices that are not visible in d$I$/d$V$ mode (see Methods for details). Fig. 3 shows d$z$/d$V$ maps acquired on the checkerboard (panels a-d) and stripes (panels j-k) lattices. Interestingly, the modes are resolved very symmetric in the $x$ and $y$ directions, i.e. the number of protrusions in both directions is equivalent, even though the unit cell of the stripes lattice is highly asymmetric.

To shine more light onto the standing wave pattern, we performed numerical calculations of artificial lattices of size $8 \times 8$ using a tight-binding approach that effectively simulate d$z$/d$V$ maps (Figs. 3e-h, l-n), which are proportional to the density of states (see Methods for details). The observed modes can be described in terms of two-dimensional confinement modes with $k$-vectors $k_x = N\pi/L$ and $k_y = M\pi/L$, where $L$ is the width of the lattice. The experimentally observed modes resemble some of the calculated modes with $N = M$ (Fig. 6). However, the ex-

perimental data show a richer structure with the links connecting the very bright protrusions in $x$ and $y$ direction. Furthermore, the experimentally observed modes are gradually transforming from one mode to another with an increasing number of lobes. At certain energies some of the lobes are not spherical, but rather have an elongated shape, that splits into two with increasing voltage.

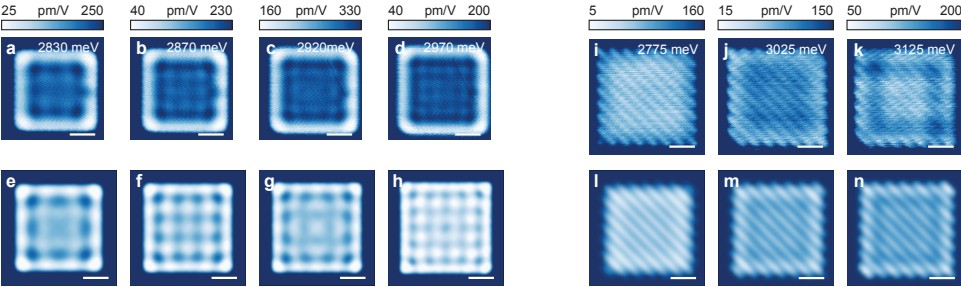

Figure 3: d$z$/d$V$ maps acquired on checkerboard and stripes lattices. (**a**, **b**, **c**, **d**) d$z$/d$V$ maps taken on an 8×8 checkerboard lattice at different energies. (**e**, **f**, **g**, **h**) Corresponding numerical calculations using a tight-binding model including an additional hybridization term. (**i**, **j**, **k**) and (**l**, **m**, **n**) Similar to (**a–d**) and (**e–h**) for an 8×8 stripes lattice. All scale bars 2 nm.

This smooth crossover can be reproduced by including the coupling of the confined modes to the electronic bath underneath. This interaction, which is mathematically represented by a self-energy term with finite imaginary part, leads to a finite lifetime of the states and a broadening of the spectral function, which consequently overlap neighbouring energy states and alter the appearance of the modes (see Methods for details on the numerical calculations). The addition of this hybridization term yields a DOS profile composed from the mixture of many individual modes that can no longer be resolved, and resembles up to very fine details the experimental STM maps, including the smooth crossover. Similar broadening of the energy modes was also attributed to strong electron-phonon coupling [24]. As such, the experimental modes have to be understood as coming from the mixture of many individual modes due to the finite coupling of the lattice to the underlying copper, so that at a particular energy we do not observe a single confined mode, but a weighted mixture of the neighbouring modes.

In a similar manner, we calculated the DOS on the stripes lattice, where the distance between vacancies along two lattice directions is not equal. We included two different hopping terms in our calculations in order to properly reproduce the experimental data. If only the hopping term along the direction of the stripes is considered, the numerically resolved modes within the chains are decoupled from each other and do not reflect the experimentally observed patterns (Figure 7). Comparison of numerical results with the experimental images allows to directly extract the effective hopping parameters for the effective square lattices. In our numerical simulations we used a model with only first neighbour hopping term of -215 meV, that reproduces the features of checkerboard lattice very well, but fails to capture the features of stripes lattice. However, the tight binding model with first and second neighbour hopping parameters -139 meV and -38 meV, respectively, can reproduce both the checkerboard and stripes lattice patterns to a great extent.

The previous discussion requires investigation of the numerical and experimental d$z$/d$V$ maps one by one and identify the interference patterns. The emergence of dispersive modes within the maps can be explored systematically using the quantitative fast Fourier transform (FFT) analysis of these d$z$/d$V$ maps images (Figs. 4a-f). In the following, we calculate the expectation value of the square of the $k$-vector $\langle k^2 \rangle$ of each FFT image, thus assigning a single value to a complete d$z$/d$V$ map (see Methods for details on how the $\langle k^2 \rangle$ values were calcu-

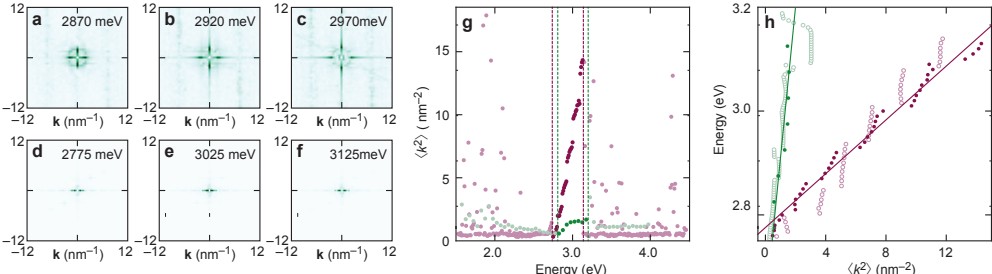

Figure 4: Fourier analysis of the d$z$/d$V$ maps. (**a**, **b**, **c**, **d**, **e**, **f**) Evolution of the Fourier map for the checkerboard lattice at different energies, corresponding to d$z$/d$V$ maps presented in Figs. 3b, c, d and i, j, k, respectively. Increasing energy leads to a shift of the maximal intensity towards higher $k$-vector values. (**g**) Evolution of the expectation value of the square of the momentum vs. the energy for checkerboard (purple) and stripes lattice (green) in a wide energy range. Dashed lines define the energy intervals within which the dispersive modes are observed in each lattice. (**h**) Dispersion plots $E$ vs. $\langle k^2 \rangle$ for the energy intervals marked in (**g**). The experimental and theory data points are represented by full and open circles, respectively. Solid lines show linear fits, from which the effective masses of experimental observed modes are extracted.

lated). Plotting the energy $E$ (i.e. the applied bias voltage) as a function of $\langle k^2 \rangle$ we provide a dispersion curve that allows to systematically identify the energy regions where an interference pattern is visible. Fig. 4h shows the obtained dispersion diagrams for the checkerboard in the energy interval 2750 meV to 3140 meV, and for the stripes lattice in the energy interval 2775 meV to 3175 meV. The full $E$ vs. $\langle k^2 \rangle$ plots are shown in Fig. 4g.

We performed linear fits to the $E$ vs. $\langle k^2 \rangle$ plots in order to extract effective electron masses for the checkerboard $m_{\mathrm{eff}} = 1.470 \pm 0.034\ m_e$ and for the stripes lattice $m_{\mathrm{eff}} = 0.131 \pm 0.025\ m_e$, where $m_e$ is the free electron mass. One would expect the stripes lattice, being highly anisotropic, to yield different effective masses for the directions parallel and perpendicular to the stripes. However, the weight in the FFT maps is found predominantly along the $k_x$ and $k_y$ axis, which are rotated 45 degrees with respect to the stripes. Therefore, a single value for the effective mass suffices to describe the observed standing wave patterns. The obtained values suggest that quasiparticle waves in the checkerboard lattice are heavier than those in the stripes lattice. While the calculations confirm this observation (theory: checkerboard: $m_{\mathrm{eff}} = 0.98 \pm 0.06\ m_e$ and stripes: $m_{\mathrm{eff}} = 0.22 \pm 0.016\ m_e$), intuitively, one might expect the checkerboard lattice, being denser than the stripes lattice, to provide greater band width due to larger hopping parameters, and therefore to yield a lower effective mass. The dispersive properties found from the analysis in Fig. 4 should therefore be considered as phenomenological only.

## Conclusion

Engineering artificial lattices by means of atom manipulation of chlorine vacancies in the Cl/Cu(100) substrate demonstrate a way to craft artificial one- and two-dimensional materials with tuneable electronic properties. We explore the emergent band formation as we build lattices of varying structure, density and size. For all lattices studied, the bottom of the emerging band is found to shift towards lower energies, in accordance to the tight-binding model, as the lattice size or density is increased. Furthermore, we find that the band onset saturates for larger structures, implying that the effect of finite size can be neglected. In the

case of two-dimensional checkerboard- and stripe shaped lattices, we observe standing Bloch waves. These patterns are well explained using a tight-binding model that includes coupling to the electron bath. Surprisingly, the effective mass of the observed Bloch waves is found to depend strongly on the lattice geometry. Our work provides a testing ground for future designer materials where the electronic properties can be defined a priori.

# Acknowledgements

J. G., F. E. K. and A. F. O. acknowledge support from the Netherlands Organisation for Scientific Research (NWO/OCW), NWO VIDI grant no. 680-47-514, and as part of the Frontiers of Nanoscience (NanoFront) program. J. L. L. and J. F.-R. acknowledge financial support by Marie-Curie-ITN 607904-SPINOGRAPH. J. F.-R. acknowledges financial support from MEC-Spain (MAT2016-78625-C2).

# Methods

**Preparation of chlorine terminated Cu(100):**   Cu(100) crystals were cleaned by repeated cycles of Argon sputtering and subsequent annealing at 550 °C. The chlorine terminated Cu(100) substrate was prepared by thermal evaporation of anhydrous $CuCl_2$ powder from a quartz crucible heated to 300 °C. Clean Cu(100) crystals are heated to 150 °C before, during and after the deposition for 10 minutes at each step [9]. The quality of the surface was verified with low-energy electron diffraction and STM.

**Acquisition of d$z$/d$V$ maps:**   The arrangement of chlorine atoms in the checkerboard lattice is very sensitive to large tunnelling currents; a current exceeding $I > 1$ μA frequently caused unintended displacement of the atoms. Such high currents are reached whilst acquiring differential conductance spectra (d$I$/d$V$), and cause the entire structure to collapse. In order to qualitatively extract the local density of states (DOS) in the checkerboard and stripes lattices, we used a method where, instead of acquiring d$I$/d$V$ spectra, we recorded the tip-sample distance $z$ as a function of applied bias voltage $V$. The time constant of the feedback-loop was much smaller (t = 25 μs) than the time set to measure a single data-point (t ∼ 1 s), ensuring thus that the tip had enough time to stabilize. In this mode the tunnelling current was kept constant at $I = 500$ pA. In the next step a numerical derivation of a $z$ vs. $V$ curve, i.e. the d$z$/d$V$ curve, has been extracted (see Fig. 5). As the tunnel current $I$ is exponentially proportional to the tip-sample distance $z$,

$$I(z) = AVe^{-2\frac{\sqrt{2m\phi}}{\hbar}z},\tag{1}$$

where $A$ is a constant, $V$ the bias voltage, $m$ the mass of the tunnelling electron, $\phi$ the height of the tunnelling barrier and $\hbar$ the reduced Planck constant. Extracting $z$ as a function of the tunnelling current and the applied bias voltage results in

$$z = \frac{\ln(\frac{I}{V}) - \ln(A)}{-2\frac{\sqrt{2m\phi}}{\hbar}}.\tag{2}$$

Derivation of the distance $z$ to voltage gives

$$\frac{dz}{dV} \propto \frac{dI}{dV}\frac{V}{I}.\tag{3}$$

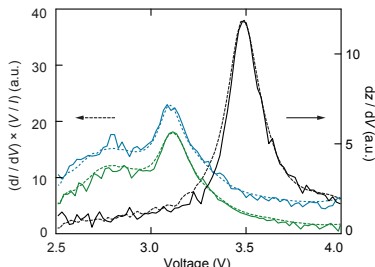

Figure 5: Comparing d$z$/d$V$ (solid lines) and d$I$/d$V$×$V$/$I$ spectra (dashed lines). Black, blue and green curves correspond respectively to the bare Cl/Cu(100) substrate and two different vacancies within the stripes lattice.

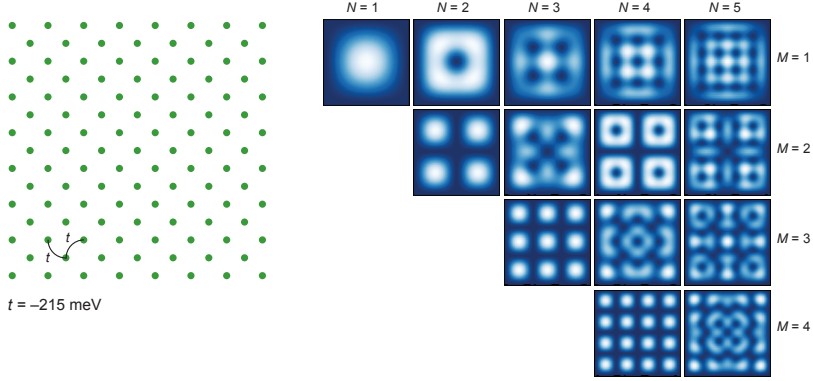

Figure 6: Numerical calculations on checkerboard lattice using a tight-binding model. Standing wave patterns acquired from the numerical calculations with the hybridization term set to zero. The integers $N$ and $M$ denote number of modes for $k_x$ and $k_y$ axis.

As can be seen from (3), the d$z$/d$V$ is linearly proportional to the normalized differential conductance spectra (d$I$/d$V$×$V$/$I$), which in turn is proportional to local DOS.

d$z$/d$V$ maps have been acquired in constant current mode, where consecutive topography images have been taken on the same area at different bias voltages in 10 mV intervals for checkerboard lattices and in 50 meV intervals for stripes lattices. The consecutive images have been subtracted, thus providing the height difference d$z$ for each point of the topography images for the respective voltage difference.

**Extracting $\langle k^2 \rangle$ values:** The acquired d$z$/d$V$ maps were transformed into FFT maps as those shown in Figure 4. Each pixel of the FFT image carries information about the intensity $I$, i.e. weight, of the corresponding $k$-vector value. In the next step the intensity and the corresponding $k$-vector value are squared, i.e. $I^2$ and $k^2$, respectively. The profiles along $k_x^2$ axis, i.e. $k_y^2$ = 0 and along $k_y^2$ axis, i.e. $k_x^2$ = 0 are normalized by the sum of $I^2$ leading to expectation values $\langle k_x^2 \rangle$ and $\langle k_y^2 \rangle$, respectively. The d$z$/d$V$ maps exhibit noise signal with very small real-space wavelength, corresponding to a large $k$-vector, thus we calculate the expectation values considering only 3 points left and 3 points right from the maximal $I^2$. Profiles along $k_y^2$ and $k_y^2$ axis appear identical and we thus calculate the expectation value $\langle k^2 \rangle = (\langle k_x^2 \rangle + \langle k_y^2 \rangle)/2$.

**Numerical calculations:** We performed numerical calculations for a model Hamiltonian defined on two different geometries: the checkerboard lattice and the stripes lattice. In both situations the size of the lattice we considered is exactly the same as in the experiment. The

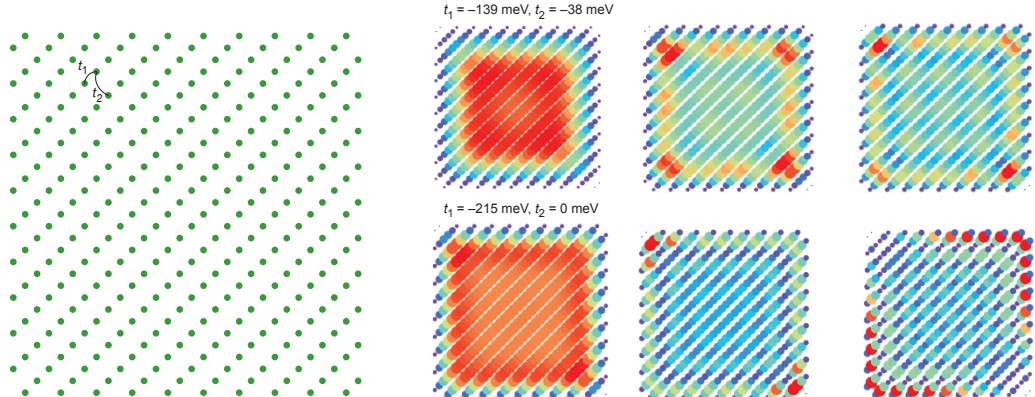

Figure 7: Numerical calculations on stripes lattice using a tight-binding model. Standing wave pattern within the stripes lattice has been simulated using nearest neighbour and next nearest neighbour hoping terms (top row) or nearest neighbour hopping term only (bottom row). Experimental data are better reproduced using both the nearest and next-nearest hopping term.

calculations are performed in the tight-binding approximation using a Hamiltonian with local orbitals in the form

$$\mathcal{H} = \sum_{ij} t_{ij} c_j^\dagger c_i, \tag{4}$$

where the parameters $t_{ij}$ are the elements of the overlap matrix between states localized within the chlorine vacancies and defined as

$$t_{ij} = \langle \psi_i | \mathcal{H} | \psi_j \rangle, \tag{5}$$

where $c_j^\dagger$ and $c_i$ are creation and annihilation operators at sites $j$ and $i$, respectively.

In our calculations, we use a value of -139 meV for first neighbour hopping term $t_1$ and a value of -38 meV for second neighbour hopping term $t_2$. Furthermore, to simulate the potential well we use the edge potential of 38 meV for the checkerboard lattice and 80 meV, for the stripes lattice.

The effect of the hybridization with the metallic bath is taken into account by means of a self-energy parameter with finite imaginary part, that enters the Dyson equation of the Green function. For simplicity we assume the self-energy term to be site-independent and diagonal, which allows to precisely reproduce the experimental features in a wide energy range. The Green function is thus defined as follows:

$$G(E) = (E - \mathcal{H} - \Sigma)^{-1}, \tag{6}$$

with self-energy term defined as

$$\Sigma = i\delta, \tag{7}$$

where $\delta = 40$ meV. Within the previous Green function, the density of states at the site $i$ is given by

$$\rho_i(E) = \mathrm{Im}(G_{ii}(E)). \tag{8}$$

The spatially resolved DOS is calculated assuming that the local state $\psi_i$ centred in $\mathbf{r}_i$ has the form

$$\psi_i(\mathbf{r}) = N e^{-(\mathbf{r}-\mathbf{r}_i)^2/\sigma^2}, \tag{9}$$

with $\sigma = 0.9d$, where $d$ is the first neighbour vacancy-vacancy distance.

Theoretical calculations of the artificial checkerboard lattice of size 8x8 using tight-binding approach with $t = -215$ meV without hybridization term are shown in Figure 6, showing the individual modes. In two dimensions, the patterns are characterized by two vectors $k_x$ and $k_y$, that are independent of each other. The vectors are defined as $k_x = N\pi/L$ and $k_y = M\pi/L$ where $N, M = (1, 2, 3, \dots , 8)$ are the mode numbers and $L$ is the size of the lattice ($L = 8$ in our case). In order to get agreement with the experiment, a finite hybridization with the metal is needed. Furthermore, in the case of the stripes lattice, an additional next nearest neighbour hopping term is needed. In Figure 7 we show calculations for the stripes lattice using the tight-binding model without hybridization term, (i) with nearest neighbour and next nearest neighbour hopping term and (ii) with nearest neighbour hopping term only. The best agreement with the experimental results is found when both terms are included.

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
