# Peer review of "Emergence of quasiparticle Bloch states in artificial crystals crafted atom-by-atom"

_SciPost Physics, doi:SciPost Phys. 2, 020 (2017)_

## Round 2 · Referee Report · Anonymous (Referee 1) · 2017-4-28

Strengths

1 - Makes an important contribution to a rapidly developing, potentially ground-breaking topic: quantum simulations of condensed matter systems.

2 - Overall, the manuscript is clearly written and the arguments are easy to follow.

Weaknesses

1 - the manuscript could/should provide some more details on various procedures.

Report

The manuscript by Jan Girovsky et al. describes the use of Cl vacancies in a monolayer of chlorine on Cu(001) to generate electronic lattices of varies geometries. To realize this, the authors cleverly combine two facts 1: there is an electronic state associated with each vacancy and 2. the position of these vacancies can be controlled using the tip of a scanning tunneling microscope. The authors exploit the atomic scale precision by which lattices can be made, to study some very fundamental aspects, such as the influence of the magnitude of the coupling between neighbors on the electronic structure. The manuscript is clearly written and overall the arguments are easy to follow.

These type of experiments can be regarded as quantum simulations of condensed matter systems. These have the potential to elucidate long-standing issues in condensed matter physics. As such, I believe this manuscript should be published, pending some minor corrections detailed below.

Requested changes

1 - On page 2, in the second paragraph, first the authors state that a monolayer of chlorine atoms on Cu(100) leads to a shift in the substrate’s work function of 1.35 eV If I am not mistaken, Figure 3b in the Supporting material of Ref 19, indicates that the shift is 1.25 eV (not 1.35 eV).

2 - On page 2, last paragraph: It may be beneficial to some readers if the authors clarify what the numbers in brackets mean.

3 - Page 2, last paragraph: Please describe how the energetic position of the conductance band minimum was determined.

4 - Page 5, second paragraph: I found it difficult to follow the arguments here. The clarity of the manuscript would be significantly improved if the authors would also show (some of) the maps simulated without surface interactions in Figure 3 (currently in Supplementary Figure 2). In addition, I suggest the authors highlight the discrepancies between the experimental maps and the maps simulated without surface interaction in the figure.

5 - Page 6, last paragraph: please describe the procedure how <k^2> was calculated.

6 - Page 6, last paragraph: Typo: 'Fig. 4e shows .....' I believe it should be 'Fig. 4f shows .....'

---

## Round 2 · Referee Report · Anonymous (Referee 2) · 2017-5-9

Strengths

  1. Paper presents a systematic fabrication of artificial lattices in atomic precision and solidifies previous work in the same system
  2. The experimental results look very reliable and indicate that defect-free lattices can be fabricated relatively routinely
  3. Simple theoretical modelling in term of a next-nearest-neighbor tight-binding model and an energy-level broadening seems to capture the essential features of observations
  4. The paper is very clearly written for most parts and

Weaknesses

  1. The artificial lattice engineering, demonstrated in a systematic manner in the present paper, has been demonstrated previously in the same system ( albeit not in as detailed manner).
  2. While the artificial lattices and their properties are demonstrated in a convincing manner, the lattice states are located much higher than the Fermi energy of the substrate. Therefore the artificial lattice remains completely unpopulated and it is not clear whether this system can be, even in principle, employed in nanoelectronics applications. This is not a shortcoming of the manuscript but rather the studied system.

Report

The manuscript of J. Girovsky and collaborators demonstrate systematic fabrication of finite 1d and 2d artificial lattices by STM methods. These systems are realized by hybridized vacancy sites on chlorine monolayer on a Cu(100) surface. The authors can explain their findings by relatively simple tight-binding model attributing one electronic orbital per vacancy site. The same system has recently been employed as a platform for topological and flatband engineering and the results of the present manuscript seem to be largely in agreement with the previous work. While the basic band-formation mechanism has been demonstrated previously, the main contribution of the manuscript is the systematic approach to different 1d and 2d lattice geometries, further solidifying the engineering of artificial lattices with atomic precision.

The manuscript is clearly written with high-quality experimental results and appropriate theoretical modeling. However, before I suggest publication of the manuscript, I would like to see clarification/modification of certain aspects of the conclusions of the paper.

Requested changes

1) It is stated that both the checkerboard and stripe lattices can be accurately modeled by the same tight-binding parameters taking into account the first and the second neighbor hopping while simple first neighbor approximation proves insufficient. Does this conclusion hold universally for the considered 1d and 2d lattices geometries and a collection of few sites? If so, then one should make a stronger case and declare that the tight-binding model with the same parameters provides a good universal description of these systems (modulo broadening which can be added by hand). If the same description is not accurate for all cases (meaning that for same separations one has to use different hopping parameters), then the reason for that should be pondered/identified.

2) Fig. 4 and the paragraph above it explains how to make the connection to the momentum dispersion of the lattice states. Especially, the authors extract the energy as a function of the average <k^2>. This information, in turn, is employed in extracting the effective mass. The lattices are anisotropic so one would expect different masses in different directions. How is this anisotropy averaged for <k^2> and how is it reflected in the FFT of dz/dV maps for stipe lattice (only checkerboard is plotted)?

3) Regarding the comparison of effective masses for stripe lattice (sl) and checkerboard lattice (cl), I do not think it is necessary “counterintuitive” that the mass of cl is higher than sl lattice. The average mass is essentially proportional to average 1/(t*a^2), where t is a hopping element and a is the lattice constant. The masses then depend on the product on t and a^2 and their size is not a priori clear from the geometry. Furthermore, it should be calculated from the tight-binding model with the extracted fitting parameters for hoppings. I would like to see the data of Fig 4 f compared to “theoretical” value from the tight-binding model with NN and NNN hopping. It could turn out that the simple comparison to the tight-binding model reproduces the values of averaged effective masses. If it does not do so, then perhaps there is something counterintuitive in the situation but before doing that there is no way to tell. I would like the authors to complement the comparison of masses with the tb model and rephrase their findings about the masses if they follow from the above simple argument.

---

## Round 2 · Referee Report · Anonymous (Referee 3) · 2017-5-15

Strengths

  1. High quality experimental data
  2. Detailed investigation of 1D and 2D lattices

Weaknesses

  1. Some detailed information missing
  2. Missing discussion of past literature
  3. Similar type of experiment has been recently reported

Report

In the manuscript by J. Girovsky et al, the authors investigate the modifications and emergence of band structure which results from coupling Cl vacancies in ordered arrays on the Cu(100). The experiments are based on low-temperature scanning tunneling microscopy and spectroscopy, and they corroborate the electronic structure of the artificial lattices with a tight-binding model. Indeed, an artificial lattice has been constructed on this system and published, as cited, but I find the results here innovative enough, especially the level of experimental detail, to merit publication. I find the experimental data of high quality, and the visualization of the electronic structure of the engineered structures very clear. The conclusions are sound and worthy of publication. I have some minor comments and questions which I detail below. The authors should address these points in a revised manuscript, before publication.

Requested changes

1. The authors fail to compare/contrast and cite the original work of N. Nilius et al, Science, 297, 1853 (2002), which was the first work of this kind. This paper originally looked at the development of 1D band structure in a nearly identical experiment. Before publication, this paper should be adequately cited and discussed in context of the new findings here.
2. Why do the authors not see the development of standing waves in the 1D chains? I find it peculiar that the length dependence of the electronic structure saturates already at six atoms? Is there some explanation for this?
3. I am missing a value of k_F, or some reference to a wavelength here? How does this compare to the length of the 1D and 2D structures?
4. On page 2, the authors write “bulk limit.” As this is not a 3D structure, I find the use of the word bulk a bit misleading. I would suggest something like long wavelength limit, or 2D limit.
5. I found it very difficult to read the color dots in Fig 1, indicating where the spectra were taken.
6. In Fig 1f, the authors use “( )” and in the paper “{ }.” I would suggest to keep this consistent, and in the text introduce what this notation means as it is just suddenly used. Can they relate {x,y} also to the crystallographic axes?
7. I fail to understand why the tight binding parameters contradict the experiment. Can the authors give more insight as to why, and in what manner this could be checked or reconciled for any future calculations for follow-up work?
8. Why is dz/dV more sensitive than dI/dV (bottom page 4)? Can this explain why the authors don’t see standing waves in the 1D structures? I’m not sure I agree with this sentence, especially if the argument is that this is just a normalized dI/dV curve?
9. A helpful suggestion: dI/dZ(V) has also been used in the past to measure band onsets. Maybe this can help in future measurements of the larger structures?

---

## Round 3 · Author Response

We thank each of the three reviewers for their thorough reading and evaluation of our manuscript, as well as for their kind and helpful suggestions to improve the text. Below, we will address all issues raised in a point-by-point manner.

---

## Round 3 · List of Changes

In response to Report 139:

1. The authors fail to compare/contrast and cite the original work of N. Nilius et al, Science, 297, 1853 (2002), which was the first work of this kind. This paper originally looked at the development of 1D band structure in a nearly identical experiment. Before publication, this paper should be adequately cited and discussed in context of the new findings here.

Authors:
We thank the reviewer for highlighting the publication, which is certainly relevant in view of the current work. The work by Nilius et al. discusses the observed modes within the free-electron model, whilst in our publication we treat the system of the coupled electronic states using the tight-binding model. In the revised version of the manuscript we cite the publication as reference 14 and add a discussion in paragraph 2. “Similar wave patterns were reported previously in assembled chains of Au atoms [14], which were best described in terms of a free electron model.” and further in paragraph 4 “The vacancy state exhibits similarities to localized states observed on gold atoms adsorbed on NiAl(110) [14], …”

2. Why do the authors not see the development of standing waves in the 1D chains? I find it peculiar that the length dependence of the electronic structure saturates already at six atoms? Is there some explanation for this?

Authors:
We speculate that the lack of standing waves in the 1D lattices is related to the relatively strong hybridization of the vacancy states with the underlying substrate. The broadened electronic states are likely to overlap with the conduction band and therefore we have not been able to resolve the confined modes experimentally. For the 2D lattices, a larger downward shift of the vacancy states was observed, resulting in less hybridization with the conduction band. Regarding the saturation of the electronic structure: as shown by Drost et. al, (Nat. Phys. 2017), the hopping integral t depends exponentially on distance. In case of sparse lattices, i.e. {3,0}, the nearest neighbours are relatively far apart and thus the interaction between adjacent vacancies is already very small and the next nearest neighbours does not contribute to that interaction. Denser lattices, having vacancies closer to each other, will have non-negligible contribution of higher order, as also demonstrated especially in the stripes and checkerboard lattices. Therefore, the point where the lattices exhibit saturation depends on the distance and the hopping term as demonstrated in Figure 1f.

3. I am missing a value of k_F, or some reference to a wavelength here? How does this compare to the length of the 1D and 2D structures?

Authors:
The effective wavelengths of the confined modes can be read from the fits in Fig. 4. All of the observed modes have k-vector smaller than the Fermi wavelength of bulk copper, k_F = 13.62 nm-1, which thus does not play a role in these modes, since the confined modes arise from hopping between localized levels. The modes seen in the experiment correspond to confined modes of the emergent lattice formed by the vacancy states. The wavelength of the pattern is determined by the wavelengths of the different modes, which show a smooth crossover as the bias is modified due to the finite coupling to the bath.

4. On page 2, the authors write “bulk limit.” As this is not a 3D structure, I find the use of the word bulk a bit misleading. I would suggest something like long wavelength limit, or 2D limit.

Authors:
We agree with the reviewer that the lattices are low dimensional and the term “bulk limit” is not appropriate. In the revised version of the manuscript we use the term “… in the limit of infinite lattice size …”, instead.

5. I found it very difficult to read the color dots in Fig 1, indicating where the spectra were taken.

Authors:
We have slightly enlarged the circles in the topography insets to make it easier to read. In view of the large number of spectra presented, the aim of the colour coding is not necessarily to match every single curve to a specific location, but rather to visualise the trend in spectroscopic evolution as a function of position inside the lattices.

6. In Fig 1f, the authors use “( )” and in the paper “{ }.” I would suggest to keep this consistent, and in the text introduce what this notation means as it is just suddenly used. Can they relate {x,y} also to the crystallographic axes?

Authors:
We thank the reviewer for noticing this discrepancy in the nomenclature and his/her suggestion to improve the consistency of the manuscript. We now use curly brackets to denote lattice spacing consistently throughout the manuscript. We also add the following lines to describe the relation of the lattice spacing with the crystallographic axis: “The notation {x,y} used here for 1D lattices describes a spacing between adjacent vacancies in the horizontal and vertical directions, respectively, in multiplies of the lattice constant a = 3.55 angstrom.” and “For 2D lattices, the notation {x,y} denotes the lattice spacing in the x and y directions in units of the lattice constant a”

7. I fail to understand why the tight binding parameters contradict the experiment. Can the authors give more insight as to why, and in what manner this could be checked or reconciled for any future calculations for follow-up work?

Authors:
An apparent smaller effective mass for the dispersive pattern in stripes than for checkerboard seems to be in conflict with a picture where smaller hopping (in one direction) yields larger effective mass. Nevertheless, the theoretical simulations of the dispersive pattern give exactly the same trend. The previous dilemma is solved by taking into account that the pattern observed in the stripes lattice not only reflects the wavelength of the confined modes, but it is also strongly influenced by the geometry of the finite stripes box. Such interplay is properly captured in the numerical simulation, but cannot be easily disentangled to extract the true effective mass associated with the tight binding parameters. Therefore, the discrepancy between the tight binding mass and the apparent effective mass in the stripes lattice comes from a geometrical effect, yet the numerical simulation of the dispersive pattern reproduce the experimental findings. We have reworded the text to reflect this notion.

8. Why is dz/dV more sensitive than dI/dV (bottom page 4)? Can this explain why the authors don’t see standing waves in the 1D structures? I’m not sure I agree with this sentence, especially if the argument is that this is just a normalized dI/dV curve?

Authors:
The referee is correct that dz/dV is just a normalized dI/dV curve, but this normalization by the total current makes it much easier to observe spectroscopic features that exist in a voltage range where the total current changes rapidly, which is the case in our measurements. For example, the dz/dV spectra taken on the stripes lattice show two peaks, whilst the dI/dV measured on very same lattice with the same tip does not reveal that.

9. A helpful suggestion: dI/dZ(V) has also been used in the past to measure band onsets. Maybe this can help in future measurements of the larger structures?

Authors:
We thank the reviewer for this helpful suggestion. We will consider implementing this into our measurement protocol for future experiments of this kind.
* * *
In response to Report 132:

1. It is stated that both the checkerboard and stripe lattices can be accurately modeled by the same tight-binding parameters taking into account the first and the second neighbor hopping while simple first neighbor approximation proves insufficient. Does this conclusion hold universally for the considered 1d and 2d lattices geometries and a collection of few sites? If so, then one should make a stronger case and declare that the tight-binding model with the same parameters provides a good universal description of these systems (modulo broadening which can be added by hand). If the same description is not accurate for all cases (meaning that for same separations one has to use different hopping parameters), then the reason for that should be pondered/identified.

Authors:
As discussed in response to Report 139, the hopping integrals depend very strongly on separation distance. For this reason, we would expect that in the case of a dense lattice, higher order neighbours need to be taken into account, whereas for sparse lattices only nearest neighbours will suffice. For this reason, we do not think it is appropriate to extend the findings for the checkerboard and stripes lattices to a universal statement.

2. Fig. 4 and the paragraph above it explains how to make the connection to the momentum dispersion of the lattice states. Especially, the authors extract the energy as a function of the average <k^2>. This information, in turn, is employed in extracting the effective mass. The lattices are anisotropic so one would expect different masses in different directions. How is this anisotropy averaged for <k^2> and how is it reflected in the FFT of dz/dV maps for stipe lattice (only checkerboard is plotted)?

Authors:
The reviewer is correct that for the stripes lattices one would in principle expect different effective masses for the directions parallel and perpendicular to the stripes. However, as the observed standing wave patterns in Fig. 3 have almost square symmetry, the corresponding weight in the FFT maps is predominantly along the kx and ky axes. These axes are 45 degrees rotated with respect to the stripes, and are therefore equivalent to each other. As such, effectively the stripes lattice is found to behave as if it has quasiparticles with isotropic dispersion. We have adjusted the text to clarify this.

3. Regarding the comparison of effective masses for stripe lattice (sl) and checkerboard lattice (cl), I do not think it is necessary “counterintuitive” that the mass of cl is higher than sl lattice. The average mass is essentially proportional to average 1/(t*a^2), where t is a hopping element and a is the lattice constant. The masses then depend on the product on t and a^2 and their size is not a priori clear from the geometry. Furthermore, it should be calculated from the tight-binding model with the extracted fitting parameters for hoppings. I would like to see the data of Fig 4 f compared to “theoretical” value from the tight-binding model with NN and NNN hopping. It could turn out that the simple comparison to the tight-binding model reproduces the values of averaged effective masses. If it does not do so, then perhaps there is something counterintuitive in the situation but before doing that there is no way to tell. I would like the authors to complement the comparison of masses with the tb model and rephrase their findings about the masses if they follow from the above simple argument.

Authors:
We followed the advice by the reviewer and found out that the simple assumption that the effective mass with the tight-binding model is proportional to 1/(t*a^2) cannot be applied here. For smaller t terms the effective mass rises and the increased distance between lattice sites is not sufficient to overcome this effect. In Figure 4f we added dispersion curves for checkerboard and stripes lattices extracted from the images simulating standing waves pattern with NN and NNN hopping terms. The effective masses extracted fitting the theoretical values are similar to the experimental ones, i.e. checkerboard effective mass from theory m_eff = 0.98 ± 0.06 m_e and stripes m_eff = 0.22 ± 0.06 m_e. We have emphasized in the manuscript that the calculations do not in fact contradict our observations.
* * *
In response to Report 124:

1. On page 2, in the second paragraph, first the authors state that a monolayer of chlorine atoms on Cu(100) leads to a shift in the substrate’s work function of 1.35 eV If I am not mistaken, Figure 3b in the Supporting material of Ref 19, indicates that the shift is 1.25 eV (not 1.35 eV).

Authors:
We thank the reviewer for noticing this typo. We have corrected it in the revised version of the manuscript.

2. On page 2, last paragraph: It may be beneficial to some readers if the authors clarify what the numbers in brackets mean.

Authors:
We have adjusted this, as described in the response to the report 139, point 6.

3. Page 2, last paragraph: Please describe how the energetic position of the conductance band minimum was determined.

Authors:
We added this information in the revised version of the manuscript. It reads as follow: “…a sharp step in the differential conductance at ~3.5 V denotes the conduction band minimum (Fig. 1a, black curve). The precise onset of the band was determined as the maximum in the normalized differential conductance dI/dV × V/I (see Fig. 5).”

4. Page 5, second paragraph: I found it difficult to follow the arguments here. The clarity of the manuscript would be significantly improved if the authors would also show (some of) the maps simulated without surface interactions in Figure 3 (currently in Supplementary Figure 2). In addition, I suggest the authors highlight the discrepancies between the experimental maps and the maps simulated without surface interaction in the figure.

Authors:
We agree that the argumentation was unclear at this point and we have revised the text to make it more understandable. In order to avoid cluttering, we would prefer not to move additional content to Fig. 3. However, we point out that the final document will contain the supplementary figures in the appendix, which will follow immediately after the text and references. What is currently supplementary figure 2 will become Fig. 6 so that the simulated maps of the bare (N,M) modes can be readily compared to the full simulations including the surface interactions.

5. Page 6, last paragraph: please describe the procedure how <k^2> was calculated.

Authors:
We add the information into the Methods section of our manuscript.

6. Page 6, last paragraph: Typo: 'Fig. 4e shows .....' I believe it should be 'Fig. 4f shows .....'

Authors:
We thank the reviewer for noticing this typo.

---

## Editorial Decision

published